# Understanding the nature of substance use in Zimbabwe: State of the art and ways forward: A scoping review protocol

**Blessing N. Marandure**[1]*, **Samson Mhizha**[2‡], **Amanda Wilson**[1‡], **Clement Nhunzvi**[3]

**1** Institute of Psychological Science, De Montfort University, Leicester, United Kingdom, **2** Department of Applied Psychology, University of Zimbabwe, Harare, Zimbabwe, **3** Department of Primary Health Care Sciences, University of Zimbabwe, Harare, Zimbabwe

◉ These authors contributed equally to this work.
‡ SM and AW also contributed equally to this work.
* nyashab@gmail.com

**Data Availability Statement:** No datasets were generated or analysed during the current study. All relevant data from this study will be made available upon study completion.

## Abstract

Reports of substance use in Zimbabwe paint a concerning picture of escalating prevalence of use, with over half of people admitted to inpatient mental health units reportedly experiencing a substance induced disorder. The country has gone through decades of significant political and socio-economical challenges, which are undoubtedly linked to the observed increases in substances use. Nevertheless, despite the resource constraints to adequately address substance use, the government has shown a renewed resolve to provide a comprehensive approach to address substance use in the country. However, there is a lack of clarity of the nature and extent of substance use and substance use disorders (SUDs), which in part is due to a lack of a national monitoring system for substance use in the country. Moreover, reports of a substance use crisis in Zimbabwe are predominantly based on anecdotal evidence, limiting the ability to gain an accurate picture of the situation. Therefore, a scoping review of the primary empirical evidence of substance use and SUDs is proposed in order to develop an adequately informed understanding of the nature of substance use and SUDs in Zimbabwe. Furthermore, the review will embed an assessment of the response to substance use, together with an analysis of the policy landscape on substance use in Zimbabwe. The PRISMA-ScR checklist will be utilised for the write up. The results of the scoping review will be essential for identifying the current state of knowledge around substance use, and identify gaps in knowledge and policy that would be a catalyst for further work to enhance knowledge and develop solutions situated within the local context. Thus the present work presents a timely effort that capitalises on current efforts by the government to address substance use in the country.

## Introduction

Substance use in Africa is on the rise, with projections estimating a 40% increase in people who use substances between 2018 to 2030 [1]. This represents the largest increase globally,

**Funding:** The authors received no external funding for this work.

**Competing interests:** The authors have declared that no competing interests exist.

with Sub-Saharan Africa projected to have the highest increase when compared to other regions in the world [2]. Moreover, East and West Africa have been identified as key players in the distribution of drugs. The identification of key drug distribution areas in Africa is concerning, particularly given the expected global increase in prevalence of substance use disorders (SUDs) due to the economic crisis brought about by the COVID-19 pandemic [1]. Additionally, projected increases in rates of substance use in Africa appear to be driven by demographic factors. Specifically, the fact that the continent's population is generally younger than other regions, together with projected sharp population rises [1]. It is thus imperative for African states, as asserted by the United Nations Office on Drugs and Crime [1] to develop sustainable and human centred approaches in response to the impending rise in substance use disorders.

This renewed call to address substance use disorders [1] comes on the back of a global focus on mental health and the desire to achieve equitable access to mental health services for mental, neurological and substance use disorders [3]. This is the key aim of the World Health Organisation's (WHO) Special Initiative for Mental Health: Universal Health Coverage for Mental Health, which was launched in 2019 [3]. One of the early adopters of this initiative from the sub-Saharan Africa region is Zimbabwe, a low to middle income country (LMIC) that has experienced significant economic, social and political challenges over the past few decades [4]. Despite these challenges, the country has shown a commitment towards strengthening its mental health systems through its National Strategic Plan for Mental Health Services (2019–2023) [5], which is in synergy with the goals of the WHO Special Initiative.

In line with data across the sub-Saharan Africa region, evidence suggests an upward trend of substance use in the Zimbabwean population [6]. For example, between 2009–2019, alcohol use was up from 9th to 8th place, in the top 10 risk factors contributing to disability adjusted life years (DALYS) in Zimbabwe [7]. Additionally, media reports paint a picture of an escalating and worrying situation of substance use, involving both licit and illicit substances. Moreover, the situation has been reported to be more complex, economic, and political, with clear signs of moving beyond teenage experimentation or chosen unhealthy behaviours [8]. Substances of concern reported on include alcohol (including illicit 'moonshine' brews), methamphetamine (crystal meth locally known as mutoriro), cough syrups containing codeine, and marijuana [9–11]. Reports of local youths identified to be in a drunken like stupor are widespread in the media, and colloquially referred to as 'ku sticker' (in reference to the paralytic like stupor youths high on substances are often found in) [11].

Anecdotal evidence also suggests a rise in substance use fuelled by the COVID-19 pandemic and resultant lockdowns [11, 12], so much so that is has been labelled an 'impending public health disaster' [6]. The ease of availability of substances, together with a lack of recreational activities for young people during lockdowns have both been cited as potential reasons for the increase [6]. However, even prior to the pandemic, drug use, particularly among the youth in Zimbabwe, was already reported to be reaching crisis levels [13], with concerns around drug use in vulnerable populations such as children living on the streets [14]. The authors assert that the significant socio-economic challenges experienced in the country are most likely linked to the observed substance use issues. Indeed, poverty is endemic in Zimbabwe, affecting 70% of the population [4], and has been identified as a risk factor for substance use [15]. Socio economic challenges are also linked to increased rates of stress, trauma, and mental health challenges which are all risk factors for substance use [15].

Strategies for addressing the identified increases in substance use are hampered by the treatment gap for mental, neurological and substance use disorders, with sub-Saharan Africa having the largest gap globally [13]. This is largely linked to the 'brain drain' due to mass emigration of mental health professionals [16], and significant underfunding for mental health services due to resource constraints [3]. For example, there is a paucity of drug and alcohol

specialist treatment facilities in Zimbabwe [3]. The situation for managing SUDs is also likely to be worsened by the emerging reports of the country becoming a hub for drug trafficking, whereby drug runners are reportedly compensated for their services using drugs, increasing availability of drugs in local communities [10, 17].

In response to both increases in substance use and lack of specialist drug treatment provision, the country recently launched the Zimbabwe National Drug Master Plan (2020–2025) [18] which aims to provide both a comprehensive and integrated approach to address the rise in substance use in the country. Within this key strategic document, the government of Zimbabwe reports that currently approximately 60% of patients admitted in mental health institutions experience substance induced disorders [18]. However, significant challenges are evident in trying to ascertain a reliable picture of the nature of SUDs in the country, owing to the evolving complexity of the problem and the lack of a national monitoring system for substance use. Subsequently, most 'evidence' cited tends to be anecdotal in nature, and heavily reliant on secondary sources. Thus, it is imperative to gain an understanding of the primary evidence and policy landscape in Zimbabwe, to gauge what is known and what is being done about substance use and SUDs. This will also aid in identifying any gaps in knowledge and assist in development of culturally and locally appropriate and sustainable solutions to addressing SUDs. With these concerns in mind, the scoping review aims to develop a broad understanding of the nature of substance use and SUDs in the Zimbabwean context. The scoping review also seeks to understand interventions that have been developed and utilised in Zimbabwe, together with an analysis of the legislative and policy landscape in relation to substance use and SUDs.

## Method

A scoping review will be conducted due to the breadth of information required to gain a broad understanding of the nature of the evidence and policy relating to substance use and SUDs in Zimbabwe. In developing the protocol, the framework developed by Arksey and O'Malley [19] together with relevant previous applications of this framework [for example, 20, 21] were consulted. Ethics approval was waived by the De Montfort University Faculty of Health and Life Sciences Research Ethics Committee.

### Stage 1: Research question

In developing the research question, the 'Population, Concept, Context' (PCC) paradigm by the Joanne Briggs Institute was utilised [22]. The 'population' is open, inclusive of both sexes and across a wide range of age groups. The 'concept' under study is substance use and use disorders, broadly encompassing both licit and illicit substances. The 'context' is geographically limited to Zimbabwe, as the review will pertain to evidence and literature from and about Zimbabwe. With these considerations in mind, the research questions are:-

1. What is the nature of substance use and SUDs in Zimbabwe?

2. What is the response to substance use and SUDs in Zimbabwe?

**Research aim.** The overarching aim of the scoping review will be to provide an assessment of the literature and policy on substance use and SUDs in Zimbabwe.

**Objectives.** The review's aim will be realised through the following objectives:

1. To identify the substances being used and/or misused within Zimbabwe

2. To understand patterns, prevalence and consequences of use, and use disorders

3.  To identify and evaluate any interventions and/or harm reduction strategies being utilised

4.  To identify and critique legislative, policy and strategic documents relevant to substance use management in Zimbabwe.

5.  To identify gaps in knowledge, policy and strategy, and provide recommendations for addressing these gaps.

### Stage 2: Identifying relevant literature

The scoping review will encompass both primary empirical studies, and policy/strategic/ legislative documents. BNM will lead on primary empirical studies and CN will lead on the policy/ strategic/legislative documents. In order to capture both of these sources, two separate successive search strategies will be employed iteratively, refining the process as necessary as the study is conducted. Empirical studies will be identified first, followed by legislative, policy and strategic documents.

**Identifying relevant studies.** The databases to be used were identified by 2 authors (BNM & CN), with the help of a subject librarian. The databases to be used for identifying primary empirical studies will be Medline (PubMed), Scopus, Academic Search Premier, the Cumulative Index to Nursing and Allied Health Literature (CINAHL Plus), Africa-Wide Information, Web of Science, PsychInfo, and PsycArticles. Grey literature sources will also be utilised in order to capture data from unpublished sources. For example, Web of Science Conference Proceedings; Grey Literature Report, and Open Grey.

In order to be included in the review, the identified studies must meet the following inclusion criteria: -

•  Studies should have been published in the last 10 years (2012–2022) in order to provide a contemporary perspective on the situation and to capture the influences of the economic & political changes post the Global Political Agreement signed between the country's two main political parties at the time [23, 24].

•  Both peer reviewed studies and grey literature will be eligible in order to comprehensively capture relevant information.

•  Studies should be empirical studies in order to capture primary data sources

•  Studies should be located in Zimbabwe and/or reporting on the Zimbabwean situation (in order to capture studies done outside but about Zimbabwe)

•  Quantitative, qualitative and mixed methodology will be included in order to represent both the nomothetic and idiographic perspectives on substance use.

In order to focus the review, the following exclusion criteria will be employed to exclude studies

•  Studies written in languages other than English or without available English translations will be excluded.

•  Animal studies will also be excluded.

**Search strategy.** Search terms were developed iteratively, and refined after running a sample web based search with broad terms for substances (substance OR drug AND Zimbabwe). Specific substances were identified from the sample search results and from the Zimbabwe

**Table 1. Search terms.**

| Keyword | *Alternatives* |
|---|---|
| Substance | Drug OR Alcohol OR [1]Musombodia OR [1]Kachasu OR Cannabis OR Marijuana OR [2]Mbanje OR Methamphetamine OR Crystal Meth OR [3]Mutoriro OR Cough Syrup OR [4]Broncleer OR Codeine OR Opioid OR Opiate OR Heroin |
| Disorder | Use OR Abuse OR Misuse OR Addiction |
| Intervention | Therapy OR Treatment OR Harm Reduction OR Rehabilitation |
| Zimbabwe | Harare OR Bulawayo |

Boolean operators (AND/ OR) will be used to combine search terms, together with truncations for capturing term variations (e.g. Addict*) as appropriate for each database.
[1]Shona colloquialisms for local moonshine alcohol brews. [2]Shona term for cannabis. [3]Shona term for crystal meth. [4] Illegally imported and banned brand of cough syrup containing codeine used locally.

National Drug Masterplan (2020–2025) [18], a key policy document developed for addressing substance use issues in Zimbabwe. This enabled the identification of substances and the associated terminology specific to the local context. See Table 1 for list of search terms.

**Identifying relevant legislative, policy and strategic documents.** Legislative documents will be identified initially from searches conducted within the Zimbabwe Legal Institute Database. References to relevant legislative, policy and strategic documents will also be followed from the empirical studies identified in the initial search. Web searches, using the Google search engine, will be conducted to find additional documents. In order to be included in the review, the documents identified must meet the following inclusion criteria: -

- The documents should specifically focus on substance use and/or SUD

- The documents should be publicly available

- Primary source outlining policy/strategy or legislation in Zimbabwe

Documents that provide a commentary on or review of substance use policy, strategy, or legislation will be excluded. This should allow us to capture the policy landscape surrounding substance use in the country, and not others' assertions on this.

## Stage 3: Study and document selection

**Study selection.** In order to select studies for inclusion in the review, multiple researchers will be employed for bias reduction. This will be carried out in different stages as follows:

1. Article titles will be reviewed in order to identify those with a key focus on substance use or SUDs. Where suitability of article is not clear it will be retained and reviewed at second stage (Research Assistant [RA] supervised by BNM)

2. Review of abstracts and titles using inclusion and exclusion criteria will be conducted by 2 simultaneous independent reviewers (BNM & RA). Cohen's K for inter-rater agreement will be calculated on the inclusion/exclusion decisions. If there is discordance between reviews, this will be resolved by a third reviewer (AW)

3. Review of full text against inclusion and exclusion criteria will be conducted by 2 simultaneous independent reviewers (BNM & CN). Cohen's K for inter-rater agreement will be calculated for the inclusion/exclusion decisions. If there is discordance between reviewers, this will be resolved by a third reviewer (SM).

**Table 2. Data extraction framework.**

| Bibliometric information | Characteristics of the study | Categories of study characteristics |
|---|---|---|
| Title | Research question/aim(s) | Definitions and conceptualisations of key concepts and associated terms: substance use and use disorder |
| Authors | Study Design | |
| Source | Sample (e.g. n =, age, geographical location [rural/urban], co-morbidity, socio-economic status/ occupation) | Patterns of use and use disorder |
| Publication Year | Type(s) of substance(s) studied | Harms associated with substance use/ use disorder |
| Profession of primary author/ academic discipline | Substance use measure utilised | Treatment Approaches |
| | Terms/definitions/conceptualisations of substance use and use disorder | |
| | Prevalence of substance use/use disorder | Summary of key message(s) |
| | Quantity/Amount of substance use | |
| | Factors influencing substance use | |
| | Consequences of substance use/use disorder | |
| | Intervention type(s) & delivery team | Areas for further development |
| | Intervention Outcome(s) | |
| | Key recommendation(s) | |

**Legislative/policy/strategic document selection.** In order to select legislative, policy and strategic documents for inclusion, full text documents will be reviewed against the inclusion criteria by 2 simultaneous independent reviewers (CN & BNM). If there is discordance between reviewers, this will be resolved by a third reviewer (SM).

## Stage 4: Charting the data

The RA will extract and chart the bibliometric information and study characteristics (under supervision by BNM) based on the following pre-created data extraction framework shown in Table 2.

Two researchers (CN & SM) will extract information from the legislative/policy/strategic documents and chart it based on the following purpose made policy checklist (Table 3). This checklist is an adaptation of a mental health systems policy checklist created by the EMBASE research consortium for LMICs, and previously applied to the Zimbabwean context [25]. Substance use specific information was also extracted from a policy checklist created by the AIDS and Rights Alliance for Southern Africa [26] and integrated within the current checklist.

## Stage 5: Collating, summarising and reporting the results

The RA will conduct a numerical analysis of bibliometric data and produce a PRISMA flow diagram for the scoping review process. A combined inductive-deductive thematic analysis of study characteristics and categories will initially be conducted by two researchers (BNM &AW). This will then be reviewed and refined by the rest of the team (CN & SM). The findings will be summarised and written up in the final report by two researchers initially (BNM &AW), with final revisions by the rest of the team (CN and SM). The Preferred Reporting Items for Systematic reviews and Meta-Analyses extension for Scoping Reviews (PRIS-MA-ScR) checklist with be utilised for writing up the scoping review [27].

For the legislative, policy and strategic documents, a combined inductive-deductive thematic analysis of study characteristics and categories will initially be conducted by two researchers (CN and SM). This will then be reviewed and refined by the rest of the team (BNM

**Table 3. Policy & strategic document checklist.**

| Bibliometric Information | Document Characteristics | Categories of document characteristics |
|---|---|---|
| Author | Consultation with service users and caregivers | Service Provision |
| | Promotes cultural approach | |
| Date Enacted | De-criminalisation of drug possession | Human Rights |
| | Specific provisions in legislation (i.e. legalisation) | |
| Description | PWID identified as key populations for HIV prevention | Special/vulnerable populations |
| | Harm Reduction (e.g. needle and syringe programmes) | Knowledge Management |
| | Specialist Drug Rehabilitation | |
| | Community Care | Interventions |
| | Evidence-based practice | |
| | Intersectoral collaboration | |
| | Finance and Funding | |
| | Prevention | |
| | Advocacy | |
| | National Monitoring System | |
| | Young people (children, adolescents, youth) | |
| | Trauma informed care | |
| | Underserved and marginalised populations (e.g. homeless) | |
| | MH Comorbidity | |

& AW). The findings will be summarised and written up in the final report by two researchers initially (CN & SM), with final revisions by the rest of the team (BNM & AW).

## Summary

The results of the scoping review will be essential for identifying the current state of knowledge around substance use, and identify gaps in knowledge and policy that would be a catalyst for further work to enhance knowledge and develop solutions situated within the local context. Thus the present work presents a timely effort that capitalises on current efforts by the government to address substance use in the country.

## Author Contributions

**Conceptualization:** Blessing N. Marandure, Samson Mhizha, Amanda Wilson, Clement Nhunzvi.

**Data curation:** Blessing N. Marandure, Clement Nhunzvi.

**Funding acquisition:** Blessing N. Marandure, Samson Mhizha, Amanda Wilson, Clement Nhunzvi.

**Methodology:** Blessing N. Marandure, Amanda Wilson, Clement Nhunzvi.

**Project administration:** Blessing N. Marandure, Clement Nhunzvi.

**Supervision:** Blessing N. Marandure, Clement Nhunzvi.

**Writing – original draft:** Blessing N. Marandure, Clement Nhunzvi.

**Writing – review & editing:** Blessing N. Marandure, Samson Mhizha, Amanda Wilson, Clement Nhunzvi.

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
