## [Decision Letter · Decision Letter 0]

31 Jan 2023

PONE-D-22-19525Understanding the nature of substance use in Zimbabwe: State of the art and ways forward: A scoping review protocolPLOS ONE

Dear Dr. Marandure,

Thank you for submitting your manuscript to PLOS ONE. After careful consideration, we feel that it has merit but does not fully meet PLOS ONE’s publication criteria as it currently stands. Therefore, we invite you to submit a revised version of the manuscript that addresses the points raised during the review process.

I encourage you to carefully consider each of the Reviewer's comments noted below. Importantly, be sure to edit the paper so that the grammar is clear, concise, and correct. Please include a numbered response to each of the Reviewer's comments in your revised manuscript.

We look forward to receiving your revised manuscript.

Kind regards,

Herb Covington

Academic Editor

PLOS ONE

Journal Requirements:

Reviewers' comments:

Reviewer's Responses to Questions

**Comments to the Author**

1. Does the manuscript provide a valid rationale for the proposed study, with clearly identified and justified research questions?

Reviewer #1: Yes

2. Is the protocol technically sound and planned in a manner that will lead to a meaningful outcome and allow testing the stated hypotheses?

Reviewer #1: Yes

3. Is the methodology feasible and described in sufficient detail to allow the work to be replicable?

Reviewer #1: Yes

4. Have the authors described where all data underlying the findings will be made available when the study is complete?

Reviewer #1: Yes

5. Is the manuscript presented in an intelligible fashion and written in standard English?

Reviewer #1: No

6. Review Comments to the Author

You may also provide optional suggestions and comments to authors that they might find helpful in planning their study.

Reviewer #1: Multiple grammatical errors. INTRO: There are statements that are made with no reference. There are connections between various concepts that are made, with no justification. Here are some examples.

No reference in line 53/54. Line 54 "This is concerning..." What is concerning? The increase in subs use in Africa vs projected highest increases in Southern Africa vs East and West Africa distributing subs?.

Line 56-58: After mentioning the rise of subs use in Africa/Southern Africa, the author fails to show reasons for this rise in subs use to now lead to "become imperative on African states... to develop..." The author also fails to make a case why they refer to the rise in substance use as a crisis.

Line 59:no ref.; also unclear how the link is made between addressing substance use disorders and achieving equitable access to mental health services.

Lines 72-74: no ref. Lines 89-90: No ref. Line 94: "the situation is further..." Author needs to clarify what situation they are referring to.

Lines: 94-99: The author fails to show the link between paucity of substance use treatment facilities in Zimbabwe and "brain drain". Is there evidence that the paucity of substance use treatment facilities is new? And possibly linked to an increasing brain drain? It is possible that Zimbabwe's paucity of subs use treatment facilities is historical, due to under-funding/other factors and predates brain drain.

Line 103: "In response to these challenges..." Again, the authors need to be clear to the reader what challenges they are referring to. Is it challenges of increasing subs use, brain drain, paucity of subs treatment facilities or emergence of drug trafficking in the country? Which of these challenges is the drug master plan a response to?

METHOD: There seems to be reference to some things done in the past vs those that will still be done. Please check the tense.

Line 122: Is Scoping Review suggested or "will be used"? I suggest that the author considers rephrasing this.

Line 206: "specific substance use focus". What does that mean? will the author exclude documents that focus on primary healthcare (PHC) systems and include substance use interventions as part of PHC?

7. PLOS authors have the option to publish the peer review history of their article (what does this mean?). If published, this will include your full peer review and any attached files.

Reviewer #1: **Yes: **Dr Tando A.S. Melapi. Department of Psychiatry; University of the Witwatersrand

---

## [Author Response · Author response to Decision Letter 0]

1 Feb 2023

We thank both the editor and the reviewer for taking the time to review our manuscript. We are pleased to be offered the opportunity to further clarify and refine our manuscript. However, we also note with concern the reviewer's response to the following question:- 5. Is the manuscript presented in an intelligible fashion and written in standard English? Whilst there were minor clarity issues identified in places, we do not feel this warrants the judgement that the manuscript is not written in an intelligible fashion and in standard English. We have highlighted the changes within the manuscript, and have provided a point by point response to the reviewer’s comments as follows:-

Reviewer #1: Multiple grammatical errors. INTRO: There are statements that are made with no reference. There are connections between various concepts that are made, with no justification. Here are some examples.

Response: We thank the reviewer for these observations. We have clarified and revised the identified issues as noted below.

Reviewer #1: No reference in line 53/54. Line 54 "This is concerning..." What is concerning? The increase in subs use in Africa vs projected highest increases in Southern Africa vs East and West Africa distributing subs?.

Response: We have now clarified the sentence’s intended meaning (line 55-56). The concern was that of the key drug distribution regions being in Africa. We believe this is a concern due to the link between distribution sites and increases in local availability of drugs identified in the latter part of the introduction.

Reviewer #1: Line 56-58: After mentioning the rise of subs use in Africa/Southern Africa, the author fails to show reasons for this rise in subs use to now lead to "become imperative on African states... to develop..." The author also fails to make a case why they refer to the rise in substance use as a crisis.

Response: The reasons for the projected rise in substance use have now been identified (lines 58-60). We also agree with the reviewer in regards to the terminology of ‘crisis’. Therefore, we have revised our assertions to be more tentative here and identified an impending rise in substance use disorders instead (line 62-63). 

Reviewer #1: Line 59:no ref.; also unclear how the link is made between addressing substance use disorders and achieving equitable access to mental health services.

Response: The statement represents the authors assertions made in relation to the call made by UNODC in the latest world drug report (referenced in lines 60-62). We have added the UNODC reference to reflect this (line 64). 

This call by UNODC was made after the launch of the WHO Special Initiative for Mental Health (lines 65-67). Hence, we assert that the call ‘comes on the back of the global focus on mental health and the desire to achieve equitable access to services (lines 64-66). We have added the WHO reference to line 66 for clarity.

The WHO Special Initiative seeks to address the treatment gap for mental, neurological and substance use disorders concurrently, and globally (and because access to these services is not equitable at present). Therefore, our assertions here are not identifying the addressing of substance use disorders in isolation as a solution to equitable access to care. 

However, considering the marginalisation of substance users in access to care more generally, it is not implausible that addressing SUDs would improve equity of access. We have not sought to expand on this as this is beyond the scope of the present paper.

Reviewer #1 Lines 72-74: no ref.

Response: The reference has now been added (now line 80). 

Reviewer #1 Lines 89-90: No ref.

Response: A reference is not required here as this represents our assertions. We have revised this to identify the statement more explicitly as the authors’ assertion (now line 99). Due to the paucity of literature, there is no primary evidence directly linking the socio-economic situation in Zimbabwe specifically to increases in substance use. What we have done instead is drawn on indirect evidence to support our assertion. Specifically, we go on to provide evidence of endemic poverty in the country (line 94) and evidence identifying poverty as a risk factor for substance use (line 96-98). Based on the identified link between poverty and substance use, it is reasonable to assert a link between the country’s socio-economic challenges and substance use.

 Reviewer #1 Line 94: "the situation is further..." Author needs to clarify what situation they are referring to.

Response: We agree, with the reviewer’s noted lack of clarity here. We have revised the statement accordingly (line 113).

Reviewer #1 Lines: 94-99: The author fails to show the link between paucity of substance use treatment facilities in Zimbabwe and "brain drain". Is there evidence that the paucity of substance use treatment facilities is new? And possibly linked to an increasing brain drain? It is possible that Zimbabwe's paucity of subs use treatment facilities is historical, due to under-funding/other factors and predates brain drain.

Response: We have identified both underfunding and the brain drain as reasons for paucity of services (lines 115-117). This is evidenced clearly in the situational analysis of the country’s healthcare system conducted as part of the WHO special initiative. Trying to unpick which came first (under-funding or brain drain) is a ‘chicken and the egg problem’ and is beyond the scope of the paper. This information is there for context of what the current challenges are:- in this case limited specialist service provision. Though we also agree that there is value in future studies to conduct in depth analysis of the reasons for the paucity of specialist services as part of efforts to remedy this.

Reviewer #1 Line 103: "In response to these challenges..." Again, the authors need to be clear to the reader what challenges they are referring to. Is it challenges of increasing subs use, brain drain, paucity of subs treatment facilities or emergence of drug trafficking in the country? Which of these challenges is the drug master plan a response to?

Response: We agree with the lack of clarity here and have identified both the increase in substance use and lack of service provision as the issues to be addressed (line 122-123).

Reviewer #1 METHOD: There seems to be reference to some things done in the past vs those that will still be done. Please check the tense.

Response: Due to the nature of a study protocol, some of the actions are future actions (i.e. the steps that will be taken to conduct the scoping review e.g line 127), whilst others are past actions (i.e. steps already taken in devising the protocol- e.g. line 129-130). Hence it is not possible for the tenses to be consistent in the method. We have reviewed the method and all tenses align with either past or present actions as they should. We have also written the protocol following the guidance from other published protocols.

Reviewer #1 Line 122: Is Scoping Review suggested or "will be used"? I suggest that the author considers rephrasing this.

Response: Thank you, we agree this was phased awkwardly. We have now amended for clarity (now line 146).

Reviewer #1 Line 206: "specific substance use focus". What does that mean? will the author exclude documents that focus on primary healthcare (PHC) systems and include substance use interventions as part of PHC?

Response: All policy and strategic documents that focus on substance use in any capacity (e.g. prevention, intervention etc) will be included in the review. This will go beyond health care systems and include wider drug policy. We have amended for clarity, but also left it broad enough to capture our intended aim (line 232).

---

## [Editor Report · Decision Letter 1]

23 Feb 2023

Understanding the nature of substance use in Zimbabwe: State of the art and ways forward: A scoping review protocol

PONE-D-22-19525R1

Dear Dr. Marandure,

We’re pleased to inform you that your manuscript has been judged scientifically suitable for publication and will be formally accepted for publication once it meets all outstanding technical requirements. I look forward to seeing your Review, on this important topic, when you have completed it. 

Kind regards,

Herb Covington

Academic Editor

PLOS ONE
---

## [Editor Report · Acceptance letter]

27 Feb 2023

PONE-D-22-19525R1 

Understanding the nature of substance use in Zimbabwe: State of the art and ways forward: A scoping review protocol 

Dear Dr. Marandure:

I'm pleased to inform you that your manuscript has been deemed suitable for publication in PLOS ONE. Congratulations! Your manuscript is now with our production department. 

Kind regards, 

on behalf of

Dr. Herb Covington 

Academic Editor

PLOS ONE